# Effects of Massive Use of Disinfectants on the Plankton Communities in Lakes from Wuhan

Gaofei Song [1], Pingping Xu [1,2], Yuxuan Zhu [1], Adilo Rediat Abate [1], Wujuan Mi [1] and Yonghong Bi [1,*]

1 State Key Laboratory of Freshwater Ecology and Biotechnology, Institute of Hydrobiology, Chinese Academy of Sciences, Wuhan 430072, China; song@ihb.ac.cn (G.S.); xupingping@ihb.ac.cn (P.X.); yxzhu@ihb.ac.cn (Y.Z.); rediat@ihb.ac.cn (A.R.A.); miwj@ihb.ac.cn (W.M.)
2 University of Chinese Academy of Sciences, Beijing 100049, China
* Correspondence: biyh@ihb.ac.cn

**Abstract:** The outbreak of COVID-19 led to the extensive use of disinfectants in urban areas. These disinfectants, along with disinfection by-products (DBPs), eventually enter waters and affected the aquatic organisms. But little information could be obtained on disinfectants threatening aquatic ecosystems. This study was conducted to obtain insight into the effects of massive use of disinfectants on freshwater ecosystems, DBPs, phytoplankton, and zooplankton in nine urban and two country lakes in Wuhan during the COVID-19 pandemic; in addition, the residual chlorine in the South Lake (one of the urban lakes), was investigated. The concentration of residual chlorine in the South Lake ranged from 0.000 mg L$^{-1}$ to 0.427mg L$^{-1}$, with an average concentration of 0.092 mg L$^{-1}$. The total concentrations of DBPs (halogenated aliphatic DBPs and aromatic halogenated DBPs) detected in the urban and country lakes ranged from 4.22 μg L$^{-1}$ to 16.59 μg L$^{-1}$ and 5.92 μg L$^{-1}$ to 7.84 μg L$^{-1}$, respectively. There was no significant difference in DBPs content between urban lakes and country lakes ($p < 0.05$). Mann–Whitney U tests showed no significant differences in plankton cell density, biomass, and alpha diversity indexes between urban and country lakes, except for the Shannon−Wiener diversity index of phytoplankton. Beta diversity demonstrated that plankton communities at different sampling stations in urban and country lakes were not significantly separated into two groups, but rather intersected each other. Variance partitioning analysis revealed that the composition of plankton communities was primarily influenced by other plankton organisms and community stability under the conditions of the investigated factors. Results indicated that the detected plankton communities in urban lakes from Wuhan were not significantly affected by the use of disinfectants. It could be deduced that the massive use of disinfectants in this outbreak had no significant impact on the plankton communities.

**Keywords:** COVID-19 pandemic; disinfectant; disinfection by-products; aquatic ecosystem; phytoplankton; zooplankton

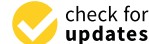



## 1. Introduction

The COVID-19 pandemic caused huge damage to human health and social economy [1–3]. Governments implemented public health protection measures, such as sanitizing public spaces, to prevent further spread of the virus [4,5]. In Wuhan alone, at least 5000 tons of chlorine-based disinfectants were used until March 2020 [6]. Chlorine was extensively employed for disinfecting domestic drinking water and wastewater in households, workplaces, public environments, and transportation during the pandemic [6–8]. It was used to minimize virus transmission through air, wastewater, and other possible routes [9–11]. However, there are potential environmental and ecological risks with the massive application of disinfectants against COVID-19 [12]. Ecologists worried that the massive use of disinfectants would pose a serious threat to aquatic ecosystems.

Some researches showed that excessive use of chlorine-based disinfectants could end up in lakes and rivers through urban drainage and surface runoff, putting aquatic

ecosystems at risk [8,13]. Residual chlorine has the potential to catalyze the oxidation of proteins, damage nucleic acids, and destroy cell walls in aquatic organisms, resulting in impacts on species diversity. Furthermore, residual disinfectants could deactivate bacteria involved in the ongoing transformation of nitrogenous compounds, disrupting the nitrogen cycle in aquatic ecosystems and converting available ammonium into the similarly toxic and more stable chloramines [13]. When residual chlorine reacts with natural organic matter, wastewater effluent organic matter, and inorganic halide ions, toxic disinfection byproducts (DBPs) could be formed, many of which display biological toxicity [14–21]. For instance, THMs and HAAs have been found to inhibit the growth of *Scenedesmus* spp., affect the swimming ability of *Daphnia magana*, and cause abnormal development or mortality in zebrafish embryos [22]. Mono-HAAs have shown cytotoxicity on HEK 293T cells [23]. Furthermore, DBPs can disrupt the relationship between other algae and *Microcystis aeruginosa*, leading to harmful cyanobacterial blooms in aquatic ecosystems [24]. Although the toxic effects of disinfectants and DBPs have been elucidated in many documents, little information could be obtained on effects of disinfectants and DBPs on aquatic ecosystem in natural waters.

Chlorinated disinfectant residues from point and nonpoint sources into natural waters poses lethal and sublethal risks to aquatic organisms. During February and March 2020, residual chlorine concentrations in some lakes in China increased up to 0.4 mg/L, which exceeds the Chinese drinking water quality standard of 0.3 mg L$^{-1}$, and were expected to have acute toxic effects on freshwater organisms [13]. After the COVID-19 pandemic outbreak, Li et al. [25] reported the presence of DBPs in the surface water of Wuhan city. These byproducts have raised concerns about ecosystem and human health safety due to their high cytotoxicity and genotoxicity [15,18,26]. Therefore, scholars appealed to local government to conduct aquatic ecological integrity assessments during and after the COVID-19 pandemic [8]. But the effects of massive use of disinfectants were unknown.

It was proved that phytoplankton and zooplankton played crucial roles in aquatic food webs, acting as primary producers and important consumers, and significantly influencing the structure and function of aquatic ecosystems [27,28]. It was increasingly evident that variations in the abundance and composition of phytoplankton and zooplankton communities were primarily driven by changes in environmental conditions [29,30]. Since different algal species display varying levels of sensitivity to disinfectants and DBPs, a certain concentration of disinfectants and DBPs could impact freshwater plankton populations [23,31]. Simultaneous analysis of different microbial groups' dynamics is crucial to understanding the responses of aquatic ecosystems to environmental changes [32]. Therefore, significant changes in the phytoplankton community can indicate whether the use of disinfectants has an impact on the aquatic ecosystem or not.

In this study, we aimed to examine the effects of disinfectant use on plankton and aquatic ecosystems. Disinfectants concentration, DBPs concentration, phytoplankton, and zooplankton from urban and country lakes were investigated. Results indicated that the massive use of disinfectants showed no significant effects on plankton communities in the urban lakes. The results helped us to obtain insight into the status of aquatic ecosystems during epidemics, and the findings serve as a basis for understanding the use of disinfectants.

## 2. Materials and Methods

### 2.1. Sample Collection

This study was carried out in 11 lakes located in Wuhan City, China: Lianhua Lake (LHH), Huanzi Lake (HZH), Liangzi Lake (LZH), Yue Lake (YH), Northwest Lake (XBH), East Lake (DH), Sha Lake (SH), Chen Lake (CH), South Lake (NH), North Lake (BH), and Moshui Lake (MSH) [33–35]. It is divided into urban lakes dominated by built-up land and country lakes dominated by grasslands and forests based on the type of land use in the vicinity [36]. Among these lakes, LHH, HZH, YH, XBH, DH, SH, NH, BH, and MSH were classified as urban lakes, while LZH and CH were categorized as country lakes. Urban

lakes are surrounded by large numbers of residents, supermarkets, etc., and conversely, country lakes are surrounded by no one. Due to the rental of sampling vessels, 7, 5 and 6 sampling stations were set up in South Lake, North Lake, and Moshui Lake, respectively, which covered the whole lake. Due to traffic control, samples of other lakes were collected on the shore, except for 4 sampling stations in East Lake, 3 sampling stations were set up in other lakes. Further information about the study lakes was provided in Figure 1 and Table S1.

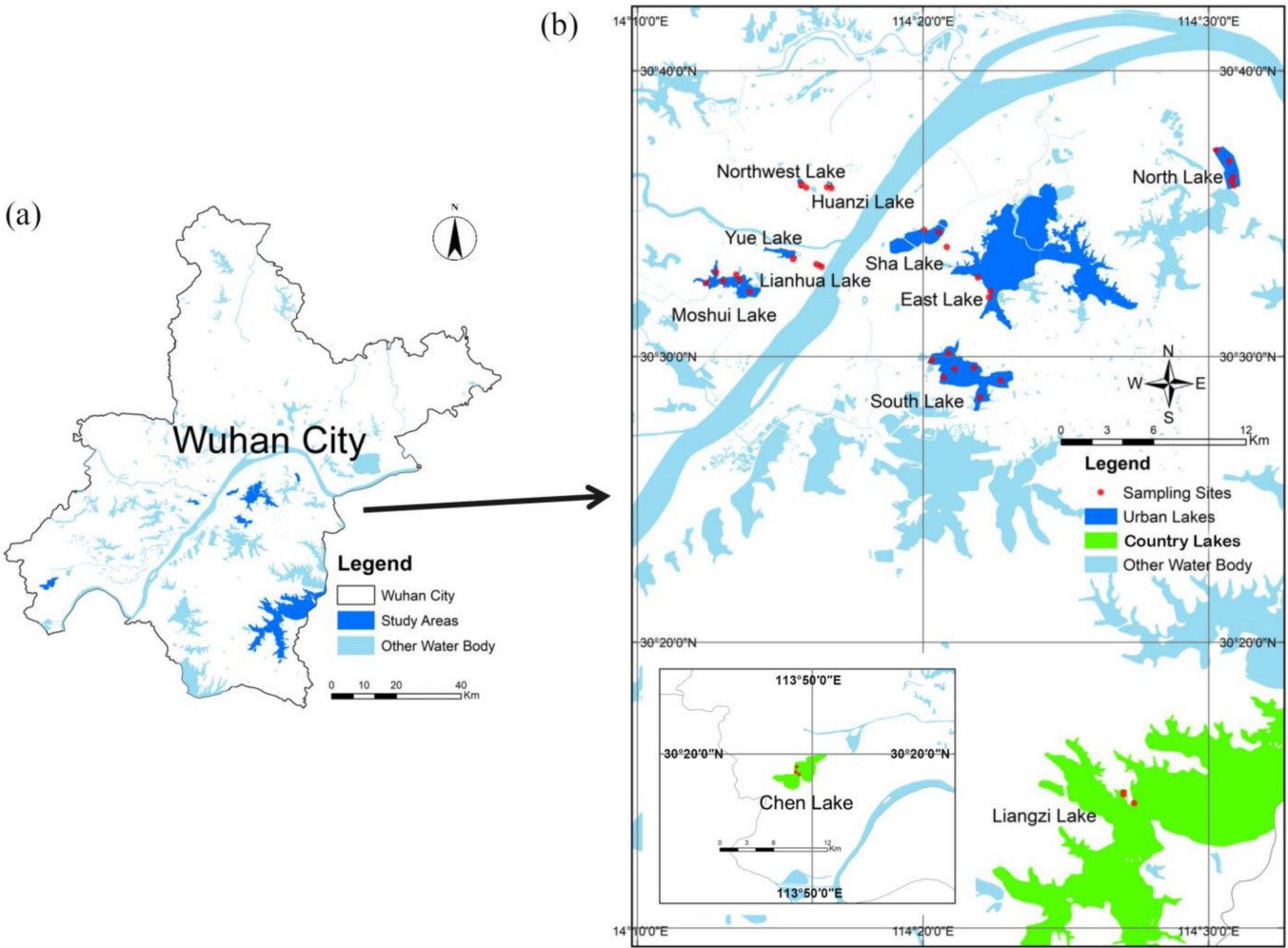

**Figure 1.** Studied area of the Wuhan City (**a**) with the sampling stations in Wuhan lakes (**b**).

The total 43 surface water samples were collected from each station between 27 April 27 and 30 April 2020. The collected water samples were subsequently divided into four subsamples: one for water chemistry analyses, and the remaining subsamples for disinfection byproducts (DBPs), phytoplankton, and zooplankton analyses, respectively. All water samples were stored in the dark at 4 °C and returned to the laboratory within two hours for further processing. For phytoplankton and microzooplankton analysis, a total of 1 L of surface water samples was fixed in situ with 1% Lugol's iodine solution and subsequently concentrated to a final volume of 30 mL. Quantitative samples of crustacean zooplankton were obtained by filtering a 20 L water sample and immediately preserving it in 4% buffered formaldehyde.

Surface water samples were collected weekly from 23 February 2020 through 29 May 2020 from seven stations (NHS1–NHS7) in South Lake, for a total of 126 samples. These samples were used to detect residual chlorine concentration.

## 2.2. Environmental Variables

Water temperature (WT), pH, dissolved oxygen (DO), electrical conductivity (SPC), and oxidation-reduction potential (ORP) were measured in situ using a Multi-parameters YSI Professional Plus instrument (YSI, Yellow Springs, OH, USA). Transparency (Trans) was measured using a Secchi disk, and turbidity (NTU) was quantified using a turbidimeter (Xinrui, WGZ-200S, Shanghai, China).

Total nitrogen (TN), ammonium ($NH_4^+$-N), nitrate ($NO_3^-$-N), total phosphorus (TP), ortho-phosphorus ($PO_4^{3-}$-P), and chlorophyll *a* (Chl *a*) were analyzed according to the standard methods of the American Public Health Association (APHA) [34]. Total carbon (TC), inorganic carbon (IC), and total organic carbon (TOC) were determined using the combustion oxidization non-dispersive infrared absorption method using a TOC analyzer (Multi N/C 3100, Jena, Germany).

As described by Li et al. [25], trihalomethanes (THMs), haloketones (HKs), haloketones (HKs), haloacetic acids (HAAs), and aromatic halogenated DBPs were detected. Halogenated aliphatic DBPs (THMs, HKs, Nas) were analyzed by gas chromatography using a triple quadrupole mass chromatograph (GC/MS-TQ8050, Shimadzu, Kyoto, Japan); and aromatic halogenated DBPs (HAAs, Aromatics) were analyzed using high performance liquid chromatography-triple quadruple mass spectrometry (HPLC/MS-TQ8060, Shimadzu, Kyoto, Japan).

The total residual chlorine concentration in South Lake was measured from 23 February through 29 May using a Q-CL501B residual chlorine and total chlorine analyzer (Shenzhen Sinsche Technology Co. Ltd., Shenzhen, China) [37]. In the laboratory, sodium hypochlorite was added to a flask containing 200 mL of South Lake water at a final concentration of 25 mg $L^{-1}$, and the concentration of sodium hypochlorite in the flask was checked at regular intervals. A pseudo-first-order model was applied to describe the kinetics of sodium hypochlorite degradation. The equation was as follows:

$$\ln([TC]/[TC]_0) = -k_{obs}t \tag{1}$$

where $k_{obs}$ means the observed pseudo-first-order rate constant ($min^{-1}$), and $[TC]_0$ and $[TC]$ represent TC concentration at time 0 and t, respectively.

## 2.3. Phytoplankton Analysis

Concentrated subsamples of 0.1 mL were mixed and identified under a microscope (Olympus, CX21, Tokyo, Japan) at 400× magnification. Phytoplankton species were identified according to Hu and Wei [38]. Each sample was analyzed using three subsamples, with at least 500 individuals counted in each subsample.

Phytoplankton cell densities in the water column were converted from microscope counts [39].

The cell volume of phytoplankton was measured and converted into biomass (1 $mm^3$ ≈ 1 mg fresh water weight) [40].

The number of species, Shannon–Wiener index, Pielou's evenness index, and Margalef species richness index, were used to analyze phytoplankton community alpha diversity [41].

## 2.4. Zooplankton Analysis

Protozoa subsamples of 0.1 mL were concentrated to 30 mL after sedimentation for 48 h and enumerated in a plankton counting chamber under a microscope at 400× magnification. Rotifera samples (1 mL) were enumerated at 100× magnification. Cladocera and Copepoda samples were separated from superplasms after sedimentation for 48 h, and the entire bottle samples were enumerated at 100× magnification using a 1 mL counting frame.

The cell density, biomass, and alpha diversity of zooplankton were calculated in the same way as the quantity analysis of phytoplankton.

*2.5. Statistical Analyses*

The relative abundance and cell density of phytoplankton and zooplankton was visualized using the "ggplot2" package in R v4.0.2. The Mann–Whitney U test was used to analyze the differences in environmental factors and plankton cell density, biomass, and alpha diversity using IBM SPSS Statistics R 26.0.0.0. Principal coordinate analysis (PCoA) based on the Bray–Curtis distance (beta diversity) was performed using the "vegan" package in R v4.0.2. Variation partitioning analysis (VPA) was used to quantify the relative effects of DBPs, the routine physics and chemistry indexes, and spatial factors (PCNM vectors) on plankton communities, as well as the relative effects of DBPs, the routine physics and chemistry indexes, and zooplankton/phytoplankton on phytoplankton/zooplankton communities [42].

## 3. Results

*3.1. Variations in Environmental Factors*

The comparison of environmental factors between urban lakes and country lakes is shown in Figure 2 and Tables 1 and 2. For the routine physics and chemistry indexes, the effects of the different types of lakes were not significant ($p > 0.05$), except for Chl *a*, SPC, pH, and ORP, which were affected by different types of lakes ($p < 0.05$) (Table 1). The average concentrations of $NO_3^--N$, $COD_{Mn}$, DO, SPC, pH, TC, IC, and TOC in urban lakes were higher compared to country lakes, whereas the average concentration of $PO_4^{3-}-P$, Turbidity, and ORP in urban lakes were lower than in country lakes.

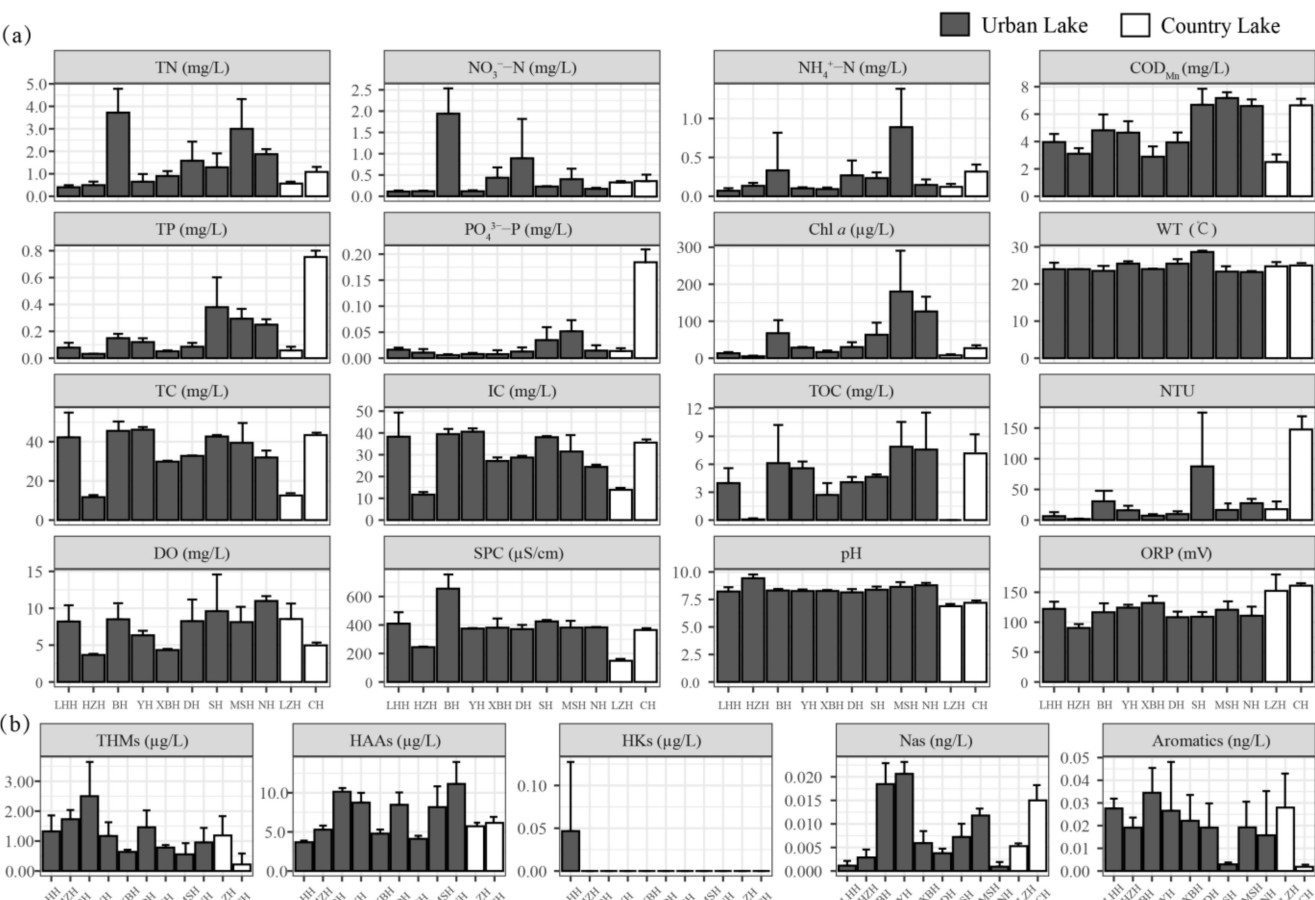

**Figure 2.** The concentrations of routine physics and chemistry factors (**a**) and DBPs (**b**) in urban lakes and country lakes.

**Table 1.** Results of the Mann–Whitney U test for the routine physics and chemistry indexes in Wuhan Lakes.

| | Urban Lakes | | | Country Lakes | | | Mann–Whitney U Test | |
|---|---|---|---|---|---|---|---|---|
| | **Avg** | **Max** | **Min** | **Avg** | **Max** | **Min** | **Z** | **p** |
| TN (mg L$^{-1}$) | 1.82 | 5.48 | 0.30 | 0.83 | 1.22 | 0.47 | −1.761 | 0.081 |
| NO$_3^-$-N (mg L$^{-1}$) | 0.54 | 2.50 | 0.09 | 0.35 | 0.46 | 0.20 | 0.935 | 0.369 |
| NH$_4^+$-N (mg L$^{-1}$) | 0.30 | 1.54 | 0.03 | 0.22 | 0.41 | 0.11 | 0.683 | 0.516 |
| TP (mg L$^{-1}$) | 0.18 | 0.62 | 0.03 | 0.40 | 0.80 | 0.03 | 0.467 | 0.661 |
| PO$_4^{3-}$-P (mg L$^{-1}$) | 0.02 | 0.09 | 0.00 | 0.10 | 0.20 | 0.00 | 1.061 | 0.297 |
| COD$_{Mn}$ (mg L$^{-1}$) | 5.20 | 8.00 | 2.23 | 4.56 | 7.09 | 2.06 | −0.935 | 0.369 |
| Turbidity (NTU) | 23.1 | 186.7 | 1.1 | 82.7 | 164.5 | 11.2 | 1.815 | 0.069 |
| Chl *a* (µg L$^{-1}$) | 75.85 | 299.02 | 3.64 | 14.32 | 34.88 | 0.94 | −2.552 | **0.009** |
| WT (°C) | 24.4 | 29.0 | 21.8 | 24.6 | 26.0 | 23.8 | 0.954 | 0.350 |
| DO (mg L$^{-1}$) | 8.0 | 15.1 | 3.5 | 6.7 | 10.4 | 4.7 | −0.935 | 0.350 |
| SPC (S m$^{-1}$) | 411.51 | 832.00 | 242.00 | 261.98 | 366.60 | 155.90 | −2.804 | **0.003** |
| pH | 8.51 | 9.64 | 7.76 | 7.04 | 7.40 | 6.74 | −3.882 | **0.000** |
| ORP (mV) | 114.8 | 145.6 | 81.5 | 157.7 | 170.6 | 120.8 | 3.379 | **0.000** |
| TC (mg L$^{-1}$) | 36.18 | 59.28 | 11.04 | 28.56 | 44.82 | 13.47 | −0.683 | 0.516 |
| IC (mg L$^{-1}$) | 30.81 | 46.72 | 10.99 | 25.46 | 37.09 | 14.65 | −1.186 | 0.250 |
| TOC (mg L$^{-1}$) | 5.37 | 15.47 | 0.00 | 3.59 | 9.31 | 0.00 | −0.809 | 0.428 |

Note: The bold font indicates a significant difference at the 0.05 level.

**Table 2.** Results of the Mann–Whitney U test for DBPs in Wuhan Lakes.

| | Urban Lakes | | | Country Lakes | | | Mann–Whitney U Test | |
|---|---|---|---|---|---|---|---|---|
| | **Avg** | **Max** | **Min** | **Avg** | **Max** | **Min** | **Z** | **p** |
| THM (µg L$^{-1}$) | 1.23 | 4.37 | 0.00 | 0.70 | 1.80 | 0.00 | −1.618 | 0.111 |
| HAA (µg L$^{-1}$) | 7.86 | 14.55 | 3.07 | 5.92 | 7.00 | 5.26 | −1.186 | 0.250 |
| HKs (µg L$^{-1}$) | 0.00 | 0.14 | 0.00 | 0.00 | 0.00 | 0.00 | −0.408 | 0.930 |
| Nas (ng L$^{-1}$) | 0.008 | 0.024 | 0.000 | 0.010 | 0.019 | 0.005 | 1.187 | 0.250 |
| Aromatics (ng L$^{-1}$) | 0.02 | 0.05 | 0.00 | 0.01 | 0.04 | 0.00 | −1.366 | 0.182 |

Note: The bold font indicates a significant difference at the 0.05 level.

Regarding DBPs, urban lakes exhibited higher concentrations of THMs, HAAs, Hks, and Aromatics, and country lakes had higher concentrations of Nas (Table 2). The concentrations of THMs (average concentration: 1.23 µg L$^{-1}$ (urban lake), 0.70 µg L$^{-1}$ (country lake)), HAAs (average concentration: 7.86 µg L$^{-1}$ (urban lake), 5.92 µg L$^{-1}$ (country lake)), HKs (average concentration: 0.004 µg L$^{-1}$ (urban lake), 0.000 µg L$^{-1}$ (country lake)), Nas (average concentration: 0.008 ng L$^{-1}$ (urban lake), 0.005 ng L$^{-1}$ (country lake)), and Aromatics (average concentration: 0.02 ng L$^{-1}$ (urban lake), 0.01 ng L$^{-1}$ (country lake)) did not significantly differ between urban lakes and country lakes (Table 2).

Total residual chlorine was detected in the water bodies of South Lake (Figure 3a), with average concentrations of 0.092 mg L$^{-1}$. Residual chlorine was not detected for most of the time at the S1 sample site. Furthermore, the S6 site had the highest total residual chlorine concentration of 0.427 mg L$^{-1}$ in February.

Figure 3b shows the degradation of sodium hypochlorite in the flask containing 200 mL of South Lake water. Approximately 75% of sodium hypochlorite was decomposed after 80 h. Sodium hypochlorite degradation fitted the pseudo-first-order kinetic model well (Equation (1)). The calculated pseudo-first-order rate constant (k$_{obs}$) was 0.0180 h$^{-1}$.

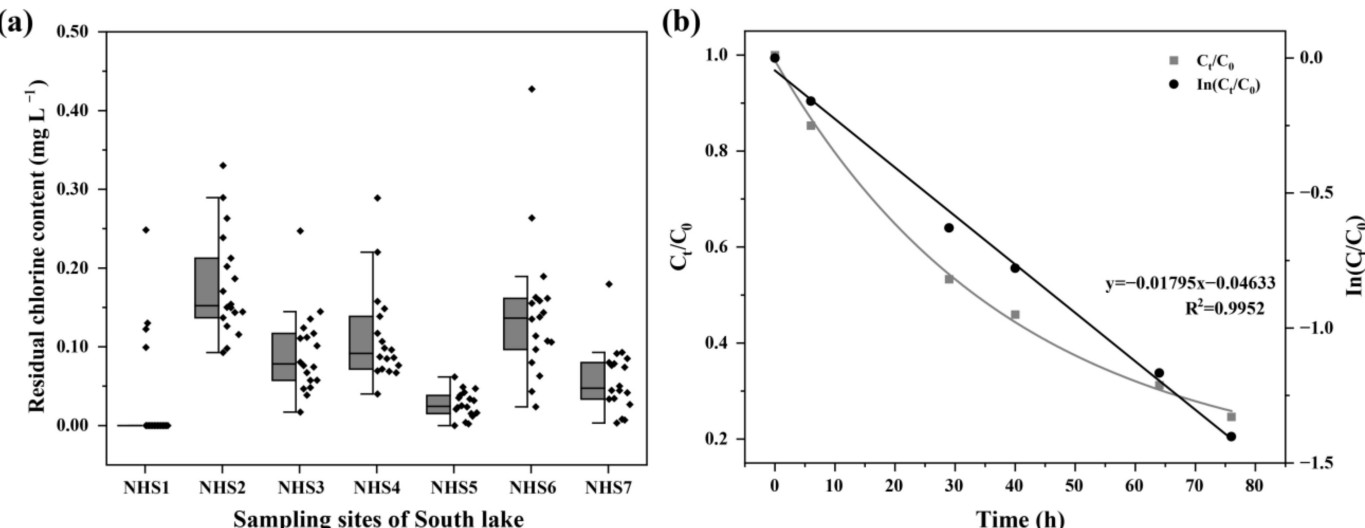

**Figure 3.** The concentrations of total residual chlorine in the South Lake (**a**) and the time-dependent decay curves and degradation kinetics of sodium hypochlorite (**b**).

### 3.2. Spatial Variation of Phytoplankton and Zooplankton Community

A total of 180 phytoplankton species (67 genera, seven phyla) and 133 zooplankton species (78 genera, four phyla) were identified across the samples. In the phytoplankton community, Chlorophyta (52.49%) emerged as the dominant phylum in the lakes of Wuhan, followed by Bacillariophyta (22.25%), Cryptophyta (11.99%), Cyanophyta (6.12%), and Euglenophyta (5.49%) (Figure 4a). In the zooplankton community, Protozoa (49.84%) and Rotifera (47.62%) were the dominant groups in the research lakes (Figure 4b).

The quantitative results indicated that the average density of phytoplankton was $9.12 \times 10^6$ cells $L^{-1}$ in urban lakes. The community structure assessment revealed mainly species of Chlorophyta (Figure 4c). In comparison, the country lakes displayed a lower cell density ($Z = -2.944$, $p = 0.053$), with the community mainly composed of Chlorophyta and Bacillariophyta. The density of zooplankton was $7.28 \times 10^3$ ind $L^{-1}$ and $7.30 \times 10^3$ ind $L^{-1}$ in urban lakes and country lakes, respectively ($Z = 0.456$, $p = 0.669$). The community structure primarily consisted of Protozoa and Rotifera (Figure 4d).

The average amounts of biomass of phytoplankton in urban and country lakes were 20.95 mg $L^{-1}$ and 15.79 mg $L^{-1}$, respectively (Figure 4e). Phytoplankton biomass was higher in urban lakes than in country lakes ($Z = -1.915$, $p = 0.056$). The average amounts of biomass of zooplankton in urban and country lakes were 9.00 mg $L^{-1}$ and 4.33 mg $L^{-1}$, respectively, and the difference was not significant ($Z = -1.507$, $p = 0.139$) (Figure 4f).

Based on the above phytoplankton and zooplankton communities, four alpha diversity indexes were calculated, as illustrated in Figure 5. In the phytoplankton community, the Mann−Whitney U test revealed that the values of the Shannon−Wiener diversity index of phytoplankton communities in urban lakes were slightly higher than those in the country lakes ($p < 0.05$). However, no significant differences were found in the species richness, Pielou's evenness index, and Margalef species richness index of the phytoplankton communities among different lake types ($p > 0.05$). Regarding zooplankton communities, no significant differences were observed in the four alpha diversity indexes between the two different types of lakes ($p > 0.05$).

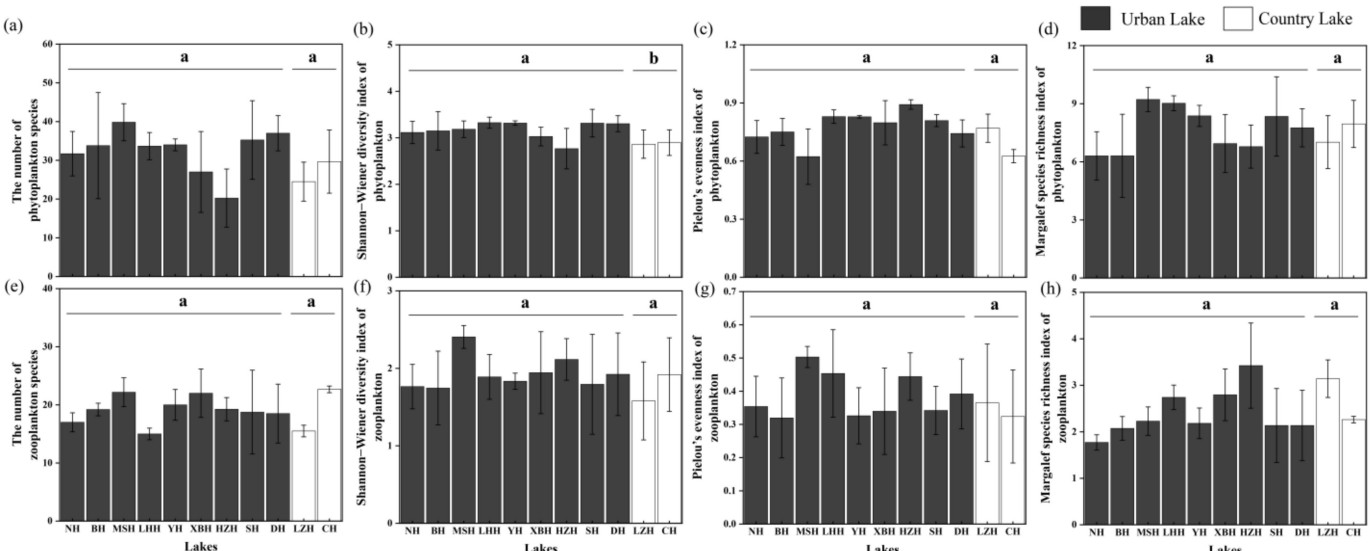

**Figure 4.** The relative abundance of phytoplankton (**a**) and zooplankton (**b**) at the phylum level. The cell density of phytoplankton (**c**) and zooplankton (**d**) at the phylum level. The biomass of phytoplankton (**e**) and zooplankton (**f**) at the phylum level.

**Figure 5.** The alpha diversity indices of phytoplankton (**a**–**d**) and zooplankton (**e**–**h**). The letters above the error bar represent statistically significant differences in the variables ($p < 0.05$).

PCoA based on Bray–Curtis distance was conducted to examine the differences in phytoplankton and/or zooplankton communities in Wuhan Lakes (Figure 6). The first two principal components (PCs) explained 21.31% and 12.12% of the total variation in the phytoplankton and zooplankton communities, respectively (Figure 6a). In the phytoplankton communities, the first two PCs explained 13.42% and 8.72% of total variation, respectively (Figure 6b), while in the zooplankton communities, the first two PCs explained 27.36% and 14.81% of the total variation, respectively (Figure 6c). The PCoA analysis revealed distinct spatial distribution patterns, with zooplankton communities in different lakes clustering into three distinct groups, whereas no obvious clustering was observed among phytoplankton communities. The results showed that the zooplankton community composition of most urban lakes was similar, but the composition of country lakes (Liangzi Lake (LZH) and Chen Lake (CH)) was more similar to Huanzi Lake (HZH), Lianhua Lake (LHH), and Moshui Lake (MSH), respectively. Plankton community composition did not cluster into two distinct groups based on the different types of urban and country lakes.

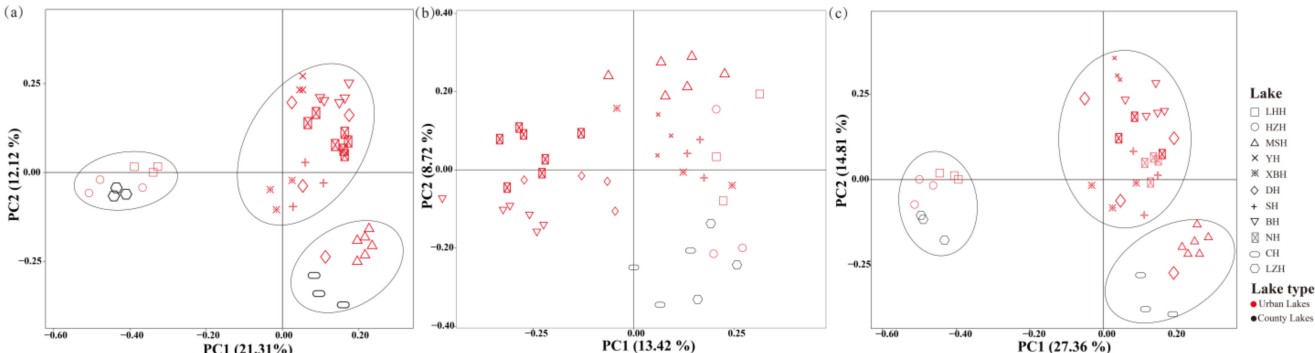

**Figure 6.** PCoA of phytoplankton and zooplankton (**a**), phytoplankton (**b**), and zooplankton (**c**) communities in Wuhan lakes.

### 3.3. Community Assembly of Phytoplankton and/or Zooplankton

To explore mechanisms underpinning the observed spatial patterns, the relative roles of different factors in community assembly were analyzed. The variables significantly associated with phytoplankton and/or zooplankton communities in each group are depicted in Figure 7. Using VPA, the conditional effects of external factors on phytoplankton and zooplankton, phytoplankton, and zooplankton communities were found to be 28.9%, 41.2%, and 87.9%, respectively. Moreover, the results of VPA showed that the DBPs explained 10.9% of the variation in the zooplankton community but only accounted for 4.9% of the variation in the phytoplankton community. These results indicated that plankton had a dominant influence on the community composition of other organisms under the conditions of the investigated factors.

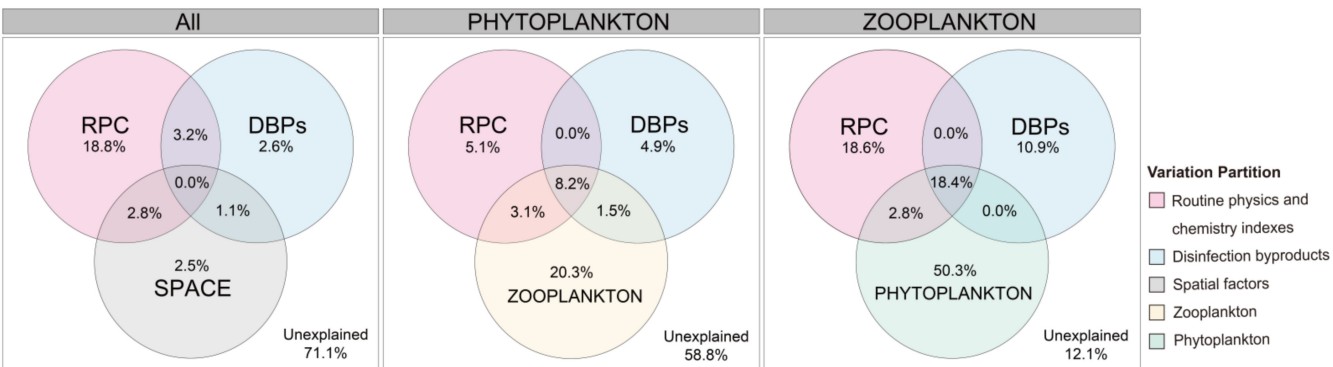

**Figure 7.** Variation partitioning analyses of contribution of different factors shaping the phytoplankton and/or zooplankton communities in Wuhan lakes.

## 4. Discussion

Due to the COVID-19 lockdown and disinfection measures, surface water resources were expected to be affected to some extent. In this study, the residual chlorine in South Lake ranged from 0.000 to 0.427 mg L$^{-1}$, with average concentrations of 0.092 mg L$^{-1}$. South Lake is a representative urban lake, and the residual chlorine level had a reference value for studying the residual chlorine content of Wuhan's urban lakes. This result was consistent with Hu's study [13], where some lakes in China had largely undetectable residual chlorine concentrations, and some lakes had residual chlorine concentrations in excess of 0.4 mg L$^{-1}$. The average concentration of residual chlorine was lower than the threshold of 0.3 mg/L (the standard for drinking water quality in China). On the other hand, residual chlorine can react with dissolved organic matter (DOM) to form DBPs, while the quantity and quality of DOM determines the type and amount of DBPs [37]. In this study, THMs and HAAs were the predominant species, accounting for over 90% of the total DBPs concentration. Before the COVID-19 pandemic, the average concentration of THMs and HAAs ranged from 0.14 to 2.8 µg L$^{-1}$ and 0.01 to 3.37 µg L$^{-1}$, respectively, in surface water in China [43]. The average concentrations of THMs in Wuhan surface water were lower than those found in Beijing and Hong Kong surface water after the pandemic. Levels of THMs and HAAs did not exceed USEPA drinking water standards either before or after the pandemic [44]. Figure 2 and Table 2 show that there was no significant difference in the content of DBPs between urban and country lakes. Considering the low residual chlorine in South Lake, despite urban lakes receiving more disinfectants than country lakes, the concentrations of DBPs showed no significant difference. Additionally, consistent with our research, chlorine disinfectant and DBPs decay rapidly in aquatic environments due to coexistence mechanisms such as volatilization, hydrolysis, adsorption, photolysis, and biodegradation [45]. This was also one of the reasons for the insignificant difference in DBPs between urban and country lakes. These findings align with current research showing that most DBPs levels after the COVID-19 outbreak remained within the range of data reported before the pandemic [46]. It was suggested that intensified disinfection did not lead to a significant increase in DBPs concentrations in the surface water of urban lakes, which may be attributed to a combination of dilution and rapid decay rates of chlorine disinfectant and DBPs in natural watersheds.

In aquatic ecosystems, the growth of plankton is closely related to water environmental factors. Changes in the water environment can lead to changes in plankton density, diversity, and community composition, which can reflect the state of the aquatic environment [47]. Our results indicated that there were no significant difference in plankton density, biomass, and alpha diversity between urban lakes and country lakes. Meanwhile, plankton community composition did not cluster into two groups based on the different types of urban and country lakes. Therefore, the massive use of disinfectants did not affect the plankton community.

In community ecology, it is important to elucidate the ecological mechanisms that control the assembly of plankton communities [42]. In this study, variance partitioning analysis further quantified the contribution of the routine physics and chemistry indexes, DBPs, and zooplankton or phytoplankton, in explaining the variation of their respective communities. The results indicate that DBPs had a low influence on community composition and plankton has a dominant influence on the community composition of other organisms. Moreover, the results showed that a large portion of variation (58.8%) in phytoplankton community structure and a small portion of variation (12.1%) in zooplankton community structure remained unexplained by extrinsic factors. These findings were consistent with previous studies conducted in different types of waters [48–51]. The higher level of unexplained variance in phytoplankton suggests that other unmeasured factors, such as bacterial interaction, herbivorous fishes, hydrology, and parasitism, may influence their dynamics [52–58]. However, Wiltshire et al. [59] compared changes in extrinsic factors potentially driving phytoplankton bloom dynamics with the changes in spring bloom phenology in the North Sea and found that the spring bloom dynamics remained relatively stable. Therefore, the most

likely explanation was that phytoplankton had evolved different aquatic survive strategies and internal feedback mechanisms that maintained a consistent community structure in a changing habitat [60,61]. Zooplankton have fewer survival strategies, resulting in lower spatial stability compared to phytoplankton. Meanwhile, the zooplankton community could be influenced by phytoplankton through a bottom-up effect. Overall, the results indicated that the community composition of plankton is primarily influenced by intrinsic community stability and interactions with other plankton organisms. Indirect evidence suggests that the use of disinfectants during the outbreak did not significantly impact the plankton in urban lakes.

## 5. Conclusions

The study examined the differences in DBPs, phytoplankton, and zooplankton in urban and country lakes during the COVID-19 epidemic when large-scale chlorine-containing disinfectant was used. The average residual chlorine concentration of 0.092 mg L$^{-1}$ in the South Lake was below the standard threshold for drinking water quality in China. There was no significant difference in DBPs content between urban and country lakes, and the DBPs levels were within the data range reported before the pandemic. Mann–Whitney U tests showed no significant differences in plankton cell density, biomass, and alpha diversity indexes between urban and country lakes. Meanwhile, beta diversity indicated that the plankton communities at different sampling stations in urban and country lakes were not significantly separated into two groups, and their community compositions were similar. Therefore, the use of disinfectants during the epidemic did not significant impact DBPs and plankton communities. Although there was no significant impact on plankton, the long term and accumulated effects on plankton and other aquatic organisms are still worthy of further investigation.

**Supplementary Materials:** The following supporting information can be downloaded at: https://www.mdpi.com/article/10.3390/w15223875/s1, Table S1: Detailed information on the sampling sites.

**Author Contributions:** Conceptualization, G.S. and Y.B.; Methodology, G.S., P.X. and A.R.A.; Software, G.S. and P.X.; Validation, Y.B.; Formal analysis, G.S. and Y.Z.; Investigation, G.S., Y.Z. and W.M.; Resources, Y.B.; Data curation, G.S. and P.X.; Writing—original draft, G.S.; Writing—review & editing, G.S., A.R.A. and Y.B.; Visualization, Y.Z.; Supervision, Y.B.; Project administration, G.S. and Y.B.; Funding acquisition, Y.B. All authors have read and agreed to the published version of the manuscript.

**Funding:** This work was funded by the National Natural Science Foundation of China (No: 31971477 and U2040210), and the Hubei Key Laboratory of Environmental and Health Effects of Persistent Toxic Substances (grant number PTS2019-07).

**Data Availability Statement:** Data are contained within the article and supplementary materials.

**Conflicts of Interest:** The authors declare no conflict of interest.

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
