# Peer review of "Effects of Massive Use of Disinfectants on the Plankton Communities in Lakes from Wuhan"

_water, doi:10.3390/w15223875_

Round 1

Reviewer 1 Report

Comments and Suggestions for Authors

In this paper, the authors present the results of an investigation into the effects of disinfectants overuse on aquatic ecosystems, DBPs, phytoplankton and zooplankton in the urban and county lakes. It is simply a very short-term monitoring study, which is standard procedure for all surface waters.

Unfortunately, the new data presented in this study are limited, and this work does not offer anything new. The fact that the data is from Wuhan during the COVID-19 pandemic does not excuse its lack of originality.

Moreover, the paper doesn't include data from similar cases around the world.

Additionally, a notable omission in this paper is the absence of Conclusions. While Conclusions are not mandatory in this journal, I believe that including them in this paper would be beneficial. Conclusions serve to summarize key findings, offer interpretation and implications, acknowledge limitations, provide recommendations, and ensure a logical flow within the paper. They play a vital role in helping both authors and readers grasp the significance of the research and its implications for the scientific community.

Despite the paper's neat presentation and well-structured writing, I suggest rejection because it lacks novelty and originality.

Reviewer 2 Report

Comments and Suggestions for Authors

The reviewed manuscript presents the results of research conducted in the initial period of the Covid-19 pandemic, i.e. between April 27 and April 30, 2020. They were conducted on 11 lakes, mainly downtown, and 2 "country" lakes. A number of physicochemical water parameters, concentrations of the remaining subsamples of Disinfection Byproducts (DBPs), and the structure of phytoplankton and zooplankton were analyzed. The aim of this study was to determine the effects of disinfectants on these plankton communities. As a result, a "photo" of the diversity of phyto- and zooplankton structure against the background of physico-chemical conditions in lakes in late spring 2020 was obtained. Nevertheless, these results can be considered interesting, which, in my opinion, were briefly well commented in the discussion. You just need to remember that, despite everything, this is a static state from the spring of 2020 and it cannot be the basis (without further research) for drawing far-reaching conclusions about the lack of DBPs impact in the long term on various communities of organisms in lakes.

Detailed notes:

Lines 15-16: Why was one urban lake (South Lake) singled out when 9 downtown lakes were studied? Why this wasn’t mentioned, i.e. how many lakes were examined and where they were located, before the results were presented?

Line 19: No significant differences were found between plankton communities, but are we talking about zoo- and phytoplankton together? However, the further content shows that differences were observed and they concerned the species richness and diversity of phyto- and zooplankton. I propose to change this fragment.

Lines 40-42: "....could have various negative effects on [....] the environment." Should it be emphasized that there are potential environmental and ecological risks? All the more so because in the cited publication this problem is actually addressed in only one sentence: "Second, because information on the ecological consequences of applying massive quantities of disinfectants in the urban environment is limited, more research on the toxic effects on urban organisms and the potential threats to the urban environment and biodiversity of this practice is urgently required.”

Lines 53-55: Sentence "[..] THMs and HAAs have been found to inhibit the growth of Scenedesmus spp., affect the swimming ability of macroalgae, and cause abnormal development and mortality in zebrafish embryos." is a verbatim quote from Liu et al. 2022 [35], which cites the results: "Cui et al. (2021) showed that THMs and HAAs inhibited the growth of Scenedesmus spp., affected the swimming ability of macroalgae, and led to abnormal development and mortality in zebrafish embryos.”

Lines 78-83: What was the purpose of introducing abbreviations of the names of the studied lakes, if they appear in the manuscript only in these lines - the definition of the abbreviation and in lines 82/83? I suggest immediately dividing the lakes into urban lakes and country lakes.

The lack of references to specific lakes in the manuscript makes the text difficult to read.

Line 83: I think it's Figure 1, not S1? Moreover, there is no information about the morphology of the studied lakes. Meanwhile, Figure 1 shows that at least two very large lakes were compared with a group of lakes, several of which were relatively small. The representativeness of the number of research points may also be problematic, as it varies in individual lakes, and the situation presented in Figure 1 does not result in any guiding idea for selecting the number and arrangement of these points. For example, in the very large Liangzi Lake - its long axis is over 40 km, only two points were located at the shore of the peninsula in the central part, while in the much smaller Lianhua Lake at least 7 points, which at least one was not located at the shore, but in the central part of the lake. The lack of data on the morphometry of the studied reservoirs and the rules for selecting the number and arrangement of points must be supplemented.

Line 130: Lack of declaration and specification of the method for counting plant plankton, or indication and description of another method to determine the relative abundance of phytoplankton.

Lines 155-156: How were the results integrated? Was the average value for the lake first calculated and then the lakes compared, or was the variability of parameters examined based on the results from all points separately? An important issue due to the different number of points in individual lakes. The text also does not mention the differences in the results obtained within individual reservoirs, i.e. whether the results were similar or significantly different within each tested reservoir.

Line 161-162: I believe that Figure S1 is very important from the point of view of the research being conducted, especially the part illustrated with Figures 5 and 6. I suggest including it in the manuscript. In lines 162-181, the average concentrations of individual compounds are compared with the limit values of the Environmental Quality Standard for Surface Water Class, completely ignoring the differences in the studied lakes, i.e. the aspect that is the subject of the summary (quoted in Figures 5 and 6). By the way, I consider this comparison to be worthless - Figure S1 shows that even two lakes differed more from each other than the other 9 urban lakes, e.g. in terms of TP concentration. What does the average TP concentration in these two lakes mean in this situation? To paraphrase - one rich man and one poor man, their average wealth provides "everyone" with a decent life, but personally I would rather be rich than poor. Or two containers of water: one with a trace concentration of harmful substances, the other with a clearly harmful concentration. Even if the average concentration of these substances does not exceed the standard for consumption, which tank would we like to drink water from?

Lines 182-185: Why is there no mention of points S1-S6 in the "Material and Methods" section? This is important when presenting results. Although Figure 2a shows one point in S6 with a given value of 0.427 mg L-1 (maximum in the tests), the figure also shows that at this point the greatest variability of the concentrations of total residual chlorine in this lake was observed, and at point S2 more often their concentration was greater than in other points.

Line 186: Important "...in natural water body".

Lines 211-213: Please improve the readability of the charts. Currently, the selected filling of the bars optically blends together and the reader cannot see anything.

Line 215-216: And why is this not mentioned in the "Material and methods" section? What was the basis for the calculations, cell number or percentage?

Lines 230-242: The description of PCoA analysis results requires major changes. The fact that the graphs explain extremely little general information in the data, and therefore obtaining any similarities within the studied zoo- and phytoplankton communities, requires some comment, or at least recognition of this fact. For example, Figure 5a shows that in three lakes LHK, HZH, LZH (may also slightly modify these abbreviations, e.g. omitting the letter H common to all lakes, probably from Chinese Hú, and avoiding non-obvious abbreviations NH for South Lake, which is right next to BH for North Lake, i.e. a reservoir with a name starting with the letter N), these communities are characterized by relatively "large" similarity, two of them are urban lakes, and one is a country lake. However, the question immediately arises for the Reader whether there were any reasons for this similarity, and here it is worth looking at Figures S1 and 3. The second question to which no attempt has been made to find an answer is the probable reason for the weaker representation of lake differences in the phytoplankton composition mentioned in lines 237-239. Can we exclude the effect of using the type of data for calculations (number or percentage) or perhaps the systematic integration of results to the phyllum level? Moreover, even the poorly legible graph 5b shows that certain tendencies towards clustering in lakes can be indicated. And this is important in the discussion, see lines 315-318.

Line 244: Please improve the readability of the charts.

Line 256-257: Sentence too general. The results show, but only in the "zooplankton" quadrant, that the characteristics of this community explained 50% of the variability in phytoplankton composition.

Reviewer 3 Report

Comments and Suggestions for Authors

The manuscript titled “Effects of massive use of disinfectants on the plankton communities in urban lakes” addresses the important and so far poorly explored topic of the impact of the mass use of disinfectants on the aquatic environment. Field work included sampling from 11 lakes at 41 sites with significant frequency (every week) at least on one of the studied lakes (the South Lake). Physico-chemical and biological analyzes of samples were performed in a wide range of parameters. The obtained database seems to be quite extensive for statistical analyses.

However, the manuscript has many shortcomings which mean that the potential of the collected information has not been fully used.

First of all, a large part of the results was presented as a comparison of "urban" and "county" lakes ecosystems. It is not clear on what criteria these two categories of water bodies (and on what purpose) were distinguished. Especially since the title only refers to "urban lakes". Moreover, the most relevant analyzes (PCoA and VPA) were performed on the combined database.

The separation of results into "urban" and "county" categories should be better justified, if the authors believe it is necessary. Maybe the division into lakes with high concentrations of DBPs and low concentrations would be more logical (taking into account other environmental factors).

In this context, the use of ANOVA analysis raises doubts. There is no equal number of observations. It would be necessary to check statistically whether the ANOVA test can be used at all or to use another test in which a different number of observations is not significant (e.g. the Mann-Whitney U test).

If I understand correctly, the analyzes of the plankton community were based on its total numbers. I would suggest more in-depth analyses, taking into account taxonomic groups and/or using plankton biomass density as an indicator (at least in the case of phytoplankton).

The manuscript did not formulate the research hypothesis or the purpose of the work.

More detailed comments.

Introduction - It is worth referring to the article: Toxicity of 17 Disinfection By-products to Different Trophic Levels of Aquatic Organisms: Ecological Risks and Mechanisms Huijun Cui, Baiyang Chen, Yuelu Jiang, Yi Tao, Xiaoshan Zhu, and Zhonghua Cai; Environmental Science & Technology 2021 55 (15), 10534-10541; DOI: 10.1021/acs.est.0c08796

Line 15 Is this higher concentration than before the COVID pandemic? Why was it assumed that it was worth investigating the impact of disinfectants on aquatic organisms?

Lines 27-28 Only the effect on plankton was tested. The conclusions cannot refer to the entire aquatic ecosystem.

Section 2.1 Is there a connection with the location of the studied sites and sewage discharge points?

Figure 1. In my opinion, the map of China is unnecessary. The map with the location of the positions should be larger.

Lines 87-88 and 98-99. How many samples were taken in total? Information about the number of samples should also be included in the explanations of statistical analyses.

Line 114 Abbreviations should be explained.

Table 1 Parameters belonging to DBPs compounds should be highlighted.

Figure 2 Is it data of South or North Lake? Different signature in the caption of the axis and in the title. Fig. 2b is incomprehensible. What is it “natural water body” ?

Fig. 3 is illegible

Line 238 It is worth marking these groups on the figure.

Figure 6 - What do you mean by environmental factors? DBPs are also environmental. Does “environmental factor” means trophy?

Line 265 Why South Lake is representative?

Round 2

Reviewer 1 Report

Comments and Suggestions for Authors

I would like to note that despite the improvements made to the manuscript, my initial assessment of the article remains unchanged. The article continues to exhibit a lack of originality and innovation, which were identified in my initial review. The revisions made in the resubmission did not sufficiently address these concerns, and as such, I am unable to recommend this work for publication. I encourage the authors to consider a more substantial revision, focusing on introducing novel and innovative elements to the research, before resubmitting it for further evaluation.

Reviewer 3 Report

Comments and Suggestions for Authors

lines 155-158, Please check the correctness of the translation. The measuring of  biomass involves measuring the volume of taxa, not measuring water density. However, we assume that the density of algae is equal to the density of water.

line 235 Why is the word urban and county written with a capital letter?
